# Damage Detection on a Beam with Multiple Cracks: A Simplified Method Based on Relative Frequency Shifts [note 1]

**DOI:** 10.3390/s21155215

**Published:** 2021-07-31

**Authors:** Gilbert-Rainer Gillich, Nuno M. M. Maia, Magd Abdel Wahab, Cristian Tufisi, Zoltan-Iosif Korka, Nicoleta Gillich, Marius Vasile Pop

**Affiliations:** 1Department of Engineering Sciences, University “Babes-Bolyai” of Cluj-Napoca, Traian Vuia Square, No. 1-4, 320085 Resita, Romania; gilbert.gillich@ubbcluj.ro (G.-R.G.); cristian.tufisi@ubbcluj.ro (C.T.); nicoleta.gillich@ubbcluj.ro (N.G.); marius.pop1@ubbcluj.ro (M.V.P.); 2IDMEC, Instituto Superior Técnico, University of Lisbon, Av. Rovisco Pais, 1049-001 Lisbon, Portugal; nuno.manuel.maia@tecnico.ulisboa.pt; 3Institute of Research and Development, Duy Tan University, 03 Quang Trung, Da Nang 550000, Vietnam; magd.abdelwahab@ugent.be; 4Laboratory Soete, Ghent University, Technologiepark Zwijnaarde 903, B-9052 Zwijnaarde, Belgium

**Keywords:** damage detection: multi-cracked beam, eigenfrequency, deflection, superposition

## Abstract

Identifying cracks in the incipient state is essential to prevent the failure of engineering structures. Detection methods relying on the analysis of the changes in modal parameters are widely used because of the advantages they present. In our previous research, we found that eigenfrequencies were capable of indicating the position and depth of damage when sufficient vibration modes were considered. The damage indicator we developed was based on the relative frequency shifts (RFS). To calculate the RFSs for various positions and depths of a crack, we established a mathematical relation that involved the squared modal curvatures in the healthy state and the deflection of the healthy and damaged beam under dead mass, respectively. In this study, we propose to calculate the RFS for beams with several cracks by applying the superposition principle. We demonstrate that this is possible if the cracks are far enough from each other. In fact, if the cracks are close to each other, the superposition method does not work and we distinguish two cases: (i) when the cracks affect the same beam face, the frequency drop is less than the sum of the individual frequency drops, and (ii) on the contrary, cracks on opposite sides cause a decrease in frequency, which is greater than the sum of the frequency drop due to individual damage. When the RFS curves are known, crack assessment becomes an optimization problem, the cost function being the distance between the measured RFSs and all possible RFSs for several vibration modes. Thus, the RFS constitutes a benchmark that characterizes damage using only the eigenfrequencies. We can accurately locate multiple cracks and estimate their severity through experiments and thus prove the reliability of the proposed method.

## 1. Introduction

The vibration-based damage detection methods have been widely used in recent decades because of the advantages they offer over traditional methods. They enable finding the location and size of the damage from vibration signals, collected from one or few measurement points not necessarily close to the damaged region. The idea on which these non-destructive testing methods are based is the relationship between the local stiffness reduction due to the crack and the modal parameter changes. Comprehensive reviews on this topic are accomplished in [1,2]. As the eigenfrequency is easily measured and implies the use of simple and robust equipment, it was the first modal parameter used in vibration-based damage detection and is still the most popular [3]. However, for small-sized damages, the capability of observing slight frequency changes is essential. Therefore, advanced signal processing is necessary when eigenfrequencies are used to detect damage [4,5].

There are two main approaches to detect damages from frequency responses. The direct problem approach presumes the existence of a set of possible damage scenarios, which include both the depth and the position of the crack. For all these scenarios the frequency responses are predicted and compared to the measured frequencies or features found involving this modal parameter. The damage is assumed to be the case for which the measured response best fits the prediction [6]. Statistical methods are most often used to compare the data [7]. Another approach is to adjust the physical-mechanical parameters of a finite element model to obtain a structural response similar to that achieved from measurements [8]. Regarding the number of cracks that can be detected, there are methods developed to identify a crack and methods that allow the identification of several cracks.

Methods to identify a single crack using eigenfrequency shifts are presented in many studies. It is demonstrated that the ratio of two relative frequency shifts achieved for two vibration modes depends only on the crack location [9]. This property is confirmed in [10] and shows that it is possible to address independently the tasks of localization and quantification of single cracks. It was successfully used to find the location of a crack [11,12].

After the location is found, the damage depth can be easily estimated [13]. Other methods to detect single cracks based on the changes in the eigenfrequencies involve different modal parameters to enrich the information regarding the structure; see, for instance, references [14,15,16].

Studies on the detection of multiple cracks in structures are also presented in the literature, but their volume is small compared to the research dedicated to detecting single crack. A typical approach in determining the modal response of beams with multiple cracks is to insert linear springs to simulate weakening due to the crack [17,18,19]. Therefore, the beam is divided into several healthy segments, the number of which depends on the number of cracks. Another current approach involves reducing the stiffness of cracked elements [20,21]. In both cases, a considerable computational effort is required to develop a database containing a relevant number of damage scenarios [22]. Finding the closest values when comparing measurements with predictions is time-consuming. Therefore, advanced optimization algorithms have been imposed, which include the genetic algorithm [23], neural networks [24], and particle swarm optimization [25].

The effect of changing environmental conditions and operational loads may affect the accuracy of the models used. For this reason, models have been developed that take these changes into account [26,27]. Because they introduce an increased degree of complexity, it is recommended to use them only if there is the certainty of variations in the condition of the beam [28,29,30,31,32].

This study is an extension of a previous conference paper [33], which considered the cracks located on opposite faces. The present research takes also in consideration cracks located on the same face of the beam. If the two closely located cracks are on the same face, the response of the beam is close to that of a beam whit a branched crack [34], which also affects a larger slice of the beam. Herein, we propose the superposition method to calculate the global frequency decrease due to multiple cracks and deduce a mathematical relation which allows calculating this decrease. Based on these findings, we extended the applicability of the previously developed damage detection method [13] for the case of multiple cracks.

Furthermore, we present a simple method to calculate the relative frequency shifts (RFS) and the eigenfrequencies of multi-crack beams. We distinguish two cases defined by the relative position of the cracks on the beam. For cracks that are quite far from each other, we show that the RFS can be calculated as the sum of the RFSs deduced for the separate cracks. If the cracks are close, their effects interfere and the principle of superposition cannot be applied. Therefore, we studied—with the help of the simulations performed employing the ANSYS program—the evolution of RFS with the distance between two cracks and we deduced the law of variation. The mathematical relationships deduced are simple and can quickly generate a comprehensive and accurate database, which can be used as a reference for detecting defects. We determine the position and depth of the cracks with an original method, which makes the correlation between the previously stated defect parameters and the RFSs. For the correct identification of defects, the database must contain as many scenarios as possible. The structure of the paper is organized as follows. Section 2 introduces the theoretical background and methodology, showing how the RFSs induced by single and multiple cracks are calculated. In Section 3 the numerical validation of the method for single and multiple cracks is presented, proofing the capability of the method to detect cracks in a real beam. Section 4 is dedicated to discussion, and conclusions are presented in Section 5.

## 2. Theoretical Background and Methodology

### 2.1. The Relative Frequency Shift (RFS) for Beams with One Crack

Damage produces a local stiffness decrease, which determines changes in the dynamic behavior. This is reflected by shifts of the eigenfrequencies and the alteration of the mode shapes and modal curvatures [1], and an increase of the damping factor [35]. The parameter that we further use in this paper to detect cracks is the eigenfrequency. To depict the frequency evolution with the crack position and severity, we design a model of the damaged beam based on the dynamic equivalence. The condition we impose is that the strain energy of the defective beam is equal to that of the intact beam. Knowing that the eigenfrequency is proportional to the stored energy, we can appreciate that the frequencies of the intact beam and the model will be equal. It should be noted here that the model makes the connection only between the crack and the eigenfrequencies of the beam, and does not provide information about the mode shapes of the damaged beam. However, we know that the deformation of the defective beam is similar to that of the healthy beam, except for an area adjacent to the crack.

In this study we consider a prismatic cantilever beam with the cross-sectional area *A = b·h* and the area moment of inertia *I* = *b·h*^3^/12. Its deflection *δ_U_* at the free end is:(1)$$\delta_{U}=\frac{\rho gAL^{4}}{\kappa EI},$$
where *ρ* is mass density, *L* is the length, *E* is the Young modulus of the beam, and *g* is the gravitational acceleration. The constant *κ* in the denominator depends on the beam’s boundary conditions. For the cantilever beam, which will be considered from now on, this constant is *κ* = 8. Note that, in Equation (1), the load is the dead mass.

According to Euler–Bernoulli’s theory, the eigenfrequencies of a healthy cantilever beam are calculated with the expression:(2)$$f_{i}^{}=\frac{\lambda_{i}^{2}}{2\pi }\sqrt{\frac{EI}{\rho AL^{4}}},$$
where *i* is the number of the bending vibration mode and *λ* is the eigenvalue. From Equations (1) and (2) we find the frequency of the first bending mode for the healthy beam as:(3)$$f_{1U}^{}=\frac{\lambda_{1}^{2}}{2\pi }\sqrt{\frac{g}{8\delta_{U}}}.$$

Let us now consider that the beam has a crack with depth *d* extended through the entire width *b*, which is located at the fixed end, where it achieves the deflection δ_(0,d)_. This happens because of the stiffness decrease in the cracked beam region that reduces the capacity of the damaged slice to store energy.

Conforming to Castigliano’s second theorem, two cantilever beams accumulate the same quantity of energy if they achieve the same deflection at the free end when the same load is applied. Therefore, we can imagine a healthy beam that achieves the same deflection *δ*_(0,*d*)_ as the cracked beam, but has a bigger mass density *δ*_(0,*d*)_ along its entire length. We nominate this beam as Equivalent Healthy Beam (EHM). Its deflection *δ*_(0,*d*)_ may be expressed in this case as:(4)$$\delta_{\left(0,d\right)}=\frac{\rho_{\left(0,d\right)}gAL^{4}}{8EI}.$$

We obtain the same results if, instead of increasing the density, we reduce the rigidity of the beam by decreasing *E* or *I*. However, it is more convenient to change the density because the relation to the deflection is more intuitive than that of Young’s modulus. On the other hand, changing the area moment of inertia requests changing the cross-sectional area and thus the beam mass is altered.

For open cracks, the deflection is similar for both extreme positions of the free end. In contrast, for the case of breathing cracks this is not true, but the time elapsed to perform one half cycle in the open stage is equal to the time elapsed to perform the other half cycle of vibration even if the performed distance is different. Hence it follows that the eigenfrequency of the beam with a breathing crack is not variable and has a unique value, at any stage the beam would be found. We have demonstrated this in [13] based on the property that in conservative systems the amount of energy is constant. Because the energy stored in the beam when achieving the extreme positions is the same, based on the proportionality between the energy and the frequency we conclude that the frequency is the same. Of course, the deflection at the free end for the two mentioned positions will be different. This justifies calculating the frequency of the beam with an open crack always using the deformation in the open stage. We also conclude that the EHB model can be used to predict beam frequencies, both in the case of open and breathing cracks.

Because of the proportionality between the frequency and the stored energy, the beam with a crack and the imagined weakened beam attain the same fundamental frequency *f*_1(0,*d*)_. It is:(5)$$f_{1(0,d)}=\frac{\lambda_{1}^{2}}{2\pi }\sqrt{\frac{EI}{\rho_{\left(0,d\right)}AL^{4}}}.$$

From Equations (4) and (5), we find the eigenfrequency of the damaged beam in respect to the free end deflection under dead mass to be:(6)$$f_{1(0,d)}=\frac{\lambda_{1}^{2}}{2\pi }\sqrt{\frac{g}{8\cdot \delta_{\left(0,d\right)}}}.$$

Thus, from Equations (3) and (6), the eigenfrequencies expression of a damaged beam results as:(7)$$f_{1(0,d)}=f_{1U}\sqrt{\frac{\delta_{U}}{\delta_{\left(0,d\right)}}}.$$

Correspondingly, the relative frequency shift $\Delta {\bar{f}}_{1(0,d)}$ can be expressed as:(8)$$\Delta {\bar{f}}_{1(0,d)}=\frac{f_{1U}-f_{1(0,d)}}{f_{1U}}=\frac{\sqrt{\delta_{\left(0,d\right)}}-\sqrt{\delta_{U}}}{\sqrt{\delta_{\left(0,d\right)}}}=\gamma (d).$$

The relative frequency shift (RFS) for the damage located at the fixed end is a measure of the damage severity and reflects the diminished stiffness of the cracked element in similar way as the torsional massless spring [36] does. We denoted the severity of transverse crack of depth *d* with *γ*(*d*). One can observe that Equations (7) and (8) no longer contain the constant associated with the beam’s end conditions and the eigenvalue because these are common factors that can be cancelled. In consequence, the severity *γ*(*d*) only characterizes the crack, being the same for any beam end conditions and bending vibration mode. We prove this in [37,38]. Unlike the methods that use fracture mechanics to determine the spring constant, the approach proposed by the authors is simple and involves only performing simulations with cracks having various depths that determine the deflection and eigenfrequencies of the beam. A method to determine an accurate mathematical relationship between the crack depth and the severity is presented in [39].

It is possible to express the frequency of the damaged beam considering Equation (7) and the severity given by Equation (8). It results in:(9)$$f_{i(0,d)}=f_{iU}-f_{iU}^{}\Delta {\bar{f}}_{i(0,d)}=f_{iU}\left[1-\frac{\sqrt{\delta_{\left(0,d\right)}}-\sqrt{\delta_{U}}}{\sqrt{\delta_{\left(0,d\right)}}}\right]=f_{iU}\left[1-\gamma (d)\right].$$

If the crack is positioned elsewhere than at the fixed end, it affects the eigenfrequencies less. It is shown in [40] that the stored energy at location *x* depends on the square of the modal curvature *ϕ_i_*″(*x*). Thus, the frequency shift and the energy lost if the same damage is located at the fixed end and at location *c*, are related in the following way:(10)$$\frac{\Delta f_{i(c,d)}}{\Delta f_{i(0,d)}}=\frac{\Delta U_{i(c,d)}}{\Delta U_{i(0,d)}}=\frac{{\left[{\phi^{\prime \prime }}_{i}\left(c\right)\right]}^{2}}{{\left[{\phi^{\prime \prime }}_{i}\left(0\right)\right]}^{2}}={\left[{\bar{\phi^{\prime \prime }}}_{i}\left(c\right)\right]}^{2}.$$

From Equations (8) and (10), the relative frequency shift due to a crack at position *c* becomes:(11)$$\Delta {\bar{f}}_{i(c,d)}=\Delta {\bar{f}}_{i(0,d)}{\left[{\bar{\phi^{\prime \prime }}}_{i}\left(c\right)\right]}^{2}=\gamma (d){\left[{\bar{\phi^{\prime \prime }}}_{i}\left(c\right)\right]}^{2}.$$

We nominate the relative frequency shift produced by a crack positioned elsewhere as at the fixed end as pseudo-severity. The dependence between severity and pseudo-severity is given by ${\left[{\bar{\phi^{\prime \prime }}}_{i}\left(c\right)\right]}^{2}$.

Taking into consideration Equation (11), the eigenfrequencies of a beam with a crack with the depth *d* which is located at a distance *c* from the fixed end are:(12)$$f_{i(c,d)}=f_{iU}-f_{iU}\Delta {\bar{f}}_{i(c,d)}=f_{iU}\left\{1-\gamma (d){\left[{\bar{\phi^{\prime \prime }}}_{i}\left(c\right)\right]}^{2}\right\}.$$

The validity of the mathematical relation deduced by the authors [41] is confirmed in [19,42]. It allows a fast and easy calculation of the eigenfrequencies for any scenario involving a single crack. Examples of relative frequency shift curves are presented in [41] for beams with four different fixing cases.

To show how the density is calculated for the EHB model we return to Castigliano’s theorem and dynamic equivalence. Figure 1 shows the model of the beam with an element having a reduced stiffness at distance *c* from the fixed end, which can be replaced by a model that has the damaged element at the fixed end but a lower stiffness decrease. Both attain the same deflection at the free end, so they have associated the same EHB model.

To ensure that EHB reaches deflection *δ*_(*c*,*d*)_ as the damaged one, we assign density *ρ*_(*c*,*d*)_ to it. So, both beams, the cracked and the EHB, store the same amount of energy and reach the same eigenfrequencies. The eigenfrequency of the damaged beam, expressed according to Equations (4) and (12), is:(13)$$f_{i(c,d)}=\frac{\lambda_{i}^{2}}{2\pi }\sqrt{\frac{EI}{\rho_{i}{}_{\left(c,d\right)}AL^{4}}}=\frac{\lambda_{i}^{2}}{2\pi }\sqrt{\frac{EI}{\rho AL^{4}}}\left\{1-\gamma (d){\left[{\bar{\phi^{\prime \prime }}}_{i}\left(c\right)\right]}^{2}\right\},$$
leading to:(14)$$\rho_{i(c,d)}=\frac{\rho }{{\left\{1-\gamma (d){\left[{\bar{\phi^{\prime \prime }}}_{i}\left(c\right)\right]}^{2}\right\}}^{2}}.$$

In consequence, when the crack is at the fixed end, the density is:(15)$$\rho_{\left(0,d\right)}=\rho \cdot {\left[\frac{f_{i}}{f_{i}(0,d)}\right]}^{2}=\frac{\rho }{{\left[1-\gamma (d)\right]}^{2}}.$$

For clarity, we use hereinafter EHB for the model that assumes a crack at the fixed end, and Mode and Location Adjusted Equivalent Healthy Beam (ML-EHB) for the model that has the crack elsewhere as at the fixed end. For the latter, the mathematical relation for the calculation of the equivalent density must include the local value of the beam curvature for the analyzed vibration mode.

### 2.2. The RFS for the Case of Beams with Multiple Cracks

If the structure has cracks in several locations, the problem of evaluating the cracks becomes much more difficult. As it is necessary to identify at least two crack positions and two depths, the methods for detecting faults are more complex and greater accuracy of the designed predictive models is required. Besides, as far as we know, there are no studies dedicated to the in-depth analysis of the behavior of beams with several defects. For this reason, there are no simple and accurate models to predict frequency changes due to multiple cracks.

In this section, we use the EHB concept and the EHB model resulting from its implementation because this model has the advantage that the second damage, if it exists, is considered to affect a healthy (equivalent) beam and thus we reduce the problem to a known one. The process can be repeated until the effect of all damages is considered. Note that the EHB model is dedicated to frequency estimation and does not reflect the changes in mode shapes due to damage.

In the previous paragraph, we assumed that the principle of superposition can be applied in the case of multiple cracks. Indeed, the deflection increase due to a crack, which depends on the additional rotation of the damaged element, is unaffected by another crack. We expect an exception in the case of two closely-located cracks, justified by the interference of their effect.

Let us consider two cracks located at distances *c*_1_ and *c*_2_, far enough from each other. The two cracks have depths *d*_1_ and *d*_2_, respectively. Based on the ML-EHB, we can define a model of the beam with the first crack as a healthy beam with higher density. The eigenfrequencies $f_{i}{}_{\left(c_{1},d_{1}\right)}$ can be found according to Equation (12) and the densities with Equation (15). Now, we can repeat the calculus by involving Equation (12) and considering that the second crack affects the healthy beam with frequency $f_{i}{}_{\left(c_{1},d_{1}\right)}$.
(16)$$f_{i}{}_{\left(c_{1},c_{2},d_{1},d_{2}\right)}=f_{i(c1,d1)}\left\{1-\gamma (d_{2}){\left[{\bar{\phi }}_{i}^{\prime \prime }(c_{2})\right]}^{2}\right\}=f_{i}\left\{1-\gamma (d_{1}){\left[{\bar{\phi }}_{i}^{\prime \prime }(c_{1})\right]}^{2}\right\}\left\{1-\gamma (d_{2}){\left[{\bar{\phi }}_{i}^{\prime \prime }(c_{2})\right]}^{2}\right\}.$$

Hence, the relative frequency shift is:(17)$$\Delta {\bar{f}}_{i}{}_{\left(c_{1},c_{2},d_{1},d_{2}\right)}=\frac{f_{i}-f_{i}{}_{\left(c_{1},c_{2},d_{1},d_{2}\right)}}{f_{i}}=\left\{1-\gamma (d_{1}){\left[{\bar{\phi }}_{i}^{\prime \prime }(c_{1})\right]}^{2}\right\}\left\{1-\gamma (d_{2}){\left[{\bar{\phi }}_{i}^{\prime \prime }(c_{2})\right]}^{2}\right\},$$
and the density results in:(18)$$\rho_{i(c1,c2d1,d2)}=\frac{\rho_{i(c1,d1)}}{{\left\{1-\gamma (d_{2}){\left[{\bar{\phi^{\prime \prime }}}_{i}\left(c_{2}\right)\right]}^{2}\right\}}^{2}}=\frac{\rho }{{\left\{1-\gamma (d_{1}){\left[{\bar{\phi^{\prime \prime }}}_{i}\left(c_{1}\right)\right]}^{2}\right\}}^{2}{\left\{1-\gamma (d_{2}){\left[{\bar{\phi^{\prime \prime }}}_{i}\left(c_{2}\right)\right]}^{2}\right\}}^{2}}.$$

Based on Equation (16), the eigenfrequencies of a beam with multiple cracks may be deduced, if the severity of the damages and their locations are known. In the case of more than 2 cracks Equation (16) becomes:(19)$$f_{i}{}_{\left(c_{1},c_{2},\ldots ,c_{m},d_{1},d_{2},d_{m}\right)}=f_{i}\prod_{j=1}^{m}\left\{1-\gamma (d_{j}){\left[{\bar{\phi }}_{i}^{\prime \prime }(c_{j})\right]}^{2}\right\},$$
where *m* is the number of cracks.

So, we can design a database with frequencies for any possible crack combinations, which is used to define the damage signatures. Obviously, the higher the number of cracks, the bigger the database. A simple and effective damage detection method that is based on this finding is introduced in the next section. It consists in finding the minimum distance between the measured/simulated frequency changes with those contained in the data base for all considered damage scenarios. For huge database, the optimization process requests the involvement of artificial intelligence [25].

### 2.3. The Proposed Damage Detection Method

Taking advantage of the fact that we can easily create a database that contains the eigenfrequencies for many damage scenarios, we have developed a damage detection method based on multimodal analysis. From the database we calculate the RFSs for several weak-axis bending vibration modes according to Equation (11). These constitute the damage patterns (DP) which are known a priori because they contain information only about the curvatures of the healthy beam and the severities dependent on the crack depths. In fact, the DP for a given damage scenario is a vector containing *n* elements that are the RFSs of the *n* vibration modes. We usually take at least five elements, so we suggest *n* ≥ 5. The bigger the *n*, the better the damage assessment results. The steps to be performed are presented in [43] and are as follows:

Calculate the first *n* weak-axis bending vibration modes for the healthy beam:(20)$$F_{U}:\left[f_{1j},\ldots \;f_{i},\ldots \;f_{n}\right],$$

Define the damage scenarios *j =* 1*…m*. It is no restriction, but these should consider 100 possible damage locations *c* on the beam, 6 damage depths *d*, and as many cracks desired.

Calculate the first *n* weak-axis bending vibration modes for the *m* damage scenarios with Equation (16):(21)$$F_{j}:\left[f_{1j},\ldots \;f_{ij},\ldots \;f_{nj}\right],$$

Calculate the RFSs for the considered scenarios with Equation (17):(22)$$\Phi_{j}:\left[\Delta {\bar{f}}_{1j},\ldots \;\Delta {\bar{f}}_{ij},\ldots \;\Delta {\bar{f}}_{nj}\right].$$

While monitoring the structure, we measure the *n* eigenfrequencies corresponding to those in the database. If changes occur, we calculate the RFSs which compose the damage signature (DS). The DS is then compared with all DP by means of an anti-distance proposed by the authors. The best fit, for which the maximum distance is found, indicates the damage scenario. The steps to be performed in situ are:

Measure the first *n* weak-axis bending vibration modes for the healthy beam:(23)$$F_{U}^{meas}:\left[f_{1U}^{meas},\ldots \;f_{iU}^{meas},\ldots \;f_{nU}^{meas}\right],$$

Measure the first n weak-axis bending vibration modes for the presumed damaged beam:(24)$$F_{D}^{meas}:\left[f_{1D}^{meas},\ldots \;f_{iD}^{meas},\ldots \;f_{nD}^{meas}\right],$$

Calculate the RFSs to find the damage signature:(25)$$\Psi :\left[\Delta {\bar{f}}_{1}^{meas},\ldots \;\Delta {\bar{f}}_{i}^{meas},\ldots \;\Delta {\bar{f}}_{n}^{meas}\right].$$

Now, the damage signature Ψ is compared with the numerous damage patterns $\Phi_{j}$ and the best fit is searched. We actually search the biggest distance between the vectors, using the relation:(26)$$nd_{j}\left(\Phi_{j}-\;\Psi \right)={\left[\sqrt{\sum_{i-1}^{n}{\left(\Phi_{ij}-\;\Psi_{i}\right)}^{2}}\right]}^{-2}.$$

Observe that *nd_j_* in Equation (26) is the Euclidean distance with a negative power, and so it becomes an anti-distance measure. The negative power causes the lowest value to increase a lot, and the rest of the values to tend to zero. The damage index is defined as:(27)$$DI=\max\left\{nd_{j}\left(\Phi_{j}-\;\Psi \right)\right\}.$$

So, we find the scenario *j* for which the damage locations *c* and depths *d* are known. The search can be performed involving intelligent optimization methods to speed up the process.

## 3. Numerical Study Regarding the Frequency Shifts Produced by Two Cracks

### 3.1. Simulation Methodology

In the following, we validate the concept of the EHB and show that the principle of superposition can be used if there is a certain distance between the transverse cracks. We also find the range in which the effect of the cracks interferes and where the principle of superposition is not applicable. A description of the frequency evolution when the cracks approach is given and we show how damage should be assessed in this case. Two cracks are involved in the tests. The first crack we generate has a fixed position on the upper face of the beam. The second crack is moved along the beam, in turn, first on the upper face, then on the lower face. We determine, by calculation using the deduced mathematical relations and by FEM analysis, the eigenfrequencies of healthy and damaged beams. The following data is obtained:-Frequencies of the healthy beam from FE analysis;-Frequencies of the beam with the fixed crack from FE analysis;-Frequencies of the beam with the fixed crack with Equation (12);-Mass density for the ML-EHB considering the beam with the fixed crack involving Equation (15);-Frequencies of the beam with two cracks by superposition, using FE analysis for the ML-EHB, on which we generate the second crack;-Frequencies of the beam with two cracks by superposition, using Equation (16);-Frequencies of the beam with two cracks from FE analysis.

The specimen we analyze is a cantilever steel beam fixed at the left end and free at the right end. Main dimensions and the physical, respective mechanical properties of the beam are shown in Table 1.

The dimensions of all cracks considered in this section are: depth *d* = 1 mm, and the distance between the transverse faces *w* = 0.001 mm. For this transverse crack, the ample study presented in [32] showed that severity is γ(1)=0.003036. The locations of the cracks are detailed in the dedicated sub-sections. Five weak-axis bending vibration modes were analyzed. The FE program involved for numerical simulations was ANSYS.

For fault detection in real structures, a simple and robust equipment described in [41] is needed. It consists of: accelerometer, acquisition board and PC with specialized software (e.g., LabView). In [5,44] it is shown how very small frequency changes can be accurately estimated.

### 3.2. Simulations Made for Cracks Located Far from Each Other

We consider here the beam with two cracks acting simultaneously on the beam. The crack on the upper face is located at distance *c_F_* = 210 mm from the fixed end, whilst the crack placed on the lower face act at locations *c_M1F_* = 100 mm, *c_M2F_* = 400 mm, *c_M3F_* = 600 mm, and *c_M4F_* = 700 mm.

We initially complete the modal analysis for one crack acting at a time and then compute the frequency drop for each damage case. The outcomes are shown in Table 2.

Thereafter, we consider the synchronal action of the upper crack and a bottom crack and performed the three modal analyses. Ultimately, we confront the frequencies obtained if two cracks act at a time with the frequency of the healthy beam from which we subtract the correspondent frequency falls. The results are shown in Table 3.

One can observe that the frequencies deduced in the two different ways agree with each other, so the superposition can be used to find the frequency drop caused by two or more cracks. This facilitates creating damage scenarios to predict the frequencies changes in the case of multiple cracks, with evident utilization in detection of damages.

Comparing the frequency results shown in Table 3 (FEM with Supeposition), one can calculate that the error between the analyzed situations does not exceed 0.005%. The error was calculated with the bellow equation:(28)$$\varepsilon =\frac{f_{FEM}-f_{Superp.}}{f_{FEM}}\cdot 100\;[\%].$$

### 3.3. Simulations Made for Closely Located Cracks

The fixed crack is located at distance *c_F_* = 210 mm from the left end of the beam and has depth *d* = 1 mm. The second crack, which also has the depth *d* = 1 mm, is located in a first scenario on the same face and, for a second scenario, on the opposite face of the beam. The positions of the second crack are fluctuating, with a step of *s* = 0.5 mm, in the range $c_{M}\in \left[200,220\right]$ mm, taken from the left end of the beam. Figure 2 provides a schematic representation of the crack positions on the beam.

In the first stage of the simulations, a healthy beam was considered. The simulations were also performed in ANSYS, by using hexahedral elements of 2 mm size, resulting a number of 251,488 elements and 1,171,495 nodes.

Afterwards, the simulation was performed on the beam having a crack in fixed position (*c_F_* = 210 mm). For both cases the deflections and the eigenfrequencies were established.

Further, using Equation (6), the severity *γ* = 0.00312 was calculated. The values of the squared modal curvatures for crack positions between the limits $c_{M}\in \left[200,220\right]$ [mm] are shown in Table 4 for the first five mode numbers. Knowing the severity and the squared modal curvature the correction coefficients were calculated. The results are also presented in Table 4.

Finally, simulations were performed for the beam having two cracks placed on the opposite faces of the beam: a fixed crack located at the distance *c_F_* = 210 mm from the left end of the beam, and a mobile crack, placed at fluctuating distances from the left end of the beam, in the range *c_M_* = 200 … 220 mm, with a step of *s* = 0.5 mm.

In order to have a global image of the influence of multiple cracks present in a cantilever beam, Figure 3 provides the evolution of the eigenfrequency for all the cases of damages presented above. Thus, the black dashed lines were drawn to show the frequencies of the undamaged beam, which noticeably are the highest values for all modes. Further, the green lines are showing the eigenfrequencies obtained by FEM for the beam with a crack that varies between the limits $c_{M}\in \left[200,220\right]$ [mm] with a step of *s* = 0.5 mm, while the hollow squares indicate the frequency determined from calculus using the squared modal curvature and severity for the same crack. Additionally, the blue lines with circles represent the eigenfrequencies obtained by FEM for the scenario where the two cracks are present on the same face, while the dashed red lines show the eigenfrequencies depicted using the superposition principle obtained from calculus using Equation (16). Finally, the lines with triangles represent the eigenfrequencies obtained by FEM for the scenario where the two cracks are present on the opposite faces of the beam.

By correlating the results plotted in Figure 3 and comparing the frequencies of the cracked beam, calculated in three different ways, one can observe that these results fit with a high precision. As a consequence, Equation (18) for appraising the eigenfrequencies of a beam with multiple cracks is validated.

## 4. Discussion

Investigating the results shown in Figure 3, we can observe that location and size prediction cannot be depicted in the case of Mode 2. These are locations where the crack produces no energy decrease and consequently it has no effect on the eigenfrequencies; this is explained from Figure 4, by the fact that the crack position for this mode number is at an inflection point, and, thus, it cannot be taken into consideration.

Analyzing how the superposition works, from the representations in Figure 4, we can conclude that in a narrow area, where the stress state due to the two cracks interferes, the actual effect on the beam frequency is different from the sum of the individual effects, resulting in the assumption that the superposition principle is valid up to a certain limit, so the frequency changes of the multiple crack beam had a similar effect to the sum of the frequency changes produced by each individual crack as long as the energy state of the damages does not interfere.

Comparing now the frequency evolution with the crack position, we observe that if the two cracks have a distance between each other bigger than 5 mm, the superposition principle can be applied.

But, if the cracks are closer by less than 5 mm, the superposition does not work because the sum of the effects of two cracks taken individually is bigger as the effect of two cracks taken at once. This happens because the effects of the two cracks interfere and these act as a single crack.

It is interesting to consider why the principle of superposition does not work if the cracks are close to each other. For this purpose, we analyzed the stress state perturbations for the crack located at *c_F_* = 210 mm*,* and the second crack was removed by *s* = 0.5 mm in the range $c_{M}\in \left[200,220\right]$ [mm].

Figure 5 shows the stress state for bending vibration mode 1 for the case when the cracks are on the same face. Areas where the stress state is disturbed by cracks are marked with a grey dotted rectangle.

If decreasing the distance between the cracks to Δ*c* < 5 mm, one can observe, starting from image c) from Figure 5, the portions on which the stress state interferes. Correlating this aspect with the frequency shifts represented in Figure 3, it can be concluded that this is the limit from which on the superposition will not work anymore. From here on, the effect of the two cracks on the frequency drop will be smaller than the sum of the effects of the two cracks.

Figure 6 presents the stress state for bending vibration mode 1 for the case when the cracks are on opposite faces. For this case, the effect of the two cracks on the frequency drop will be greater than the sum of the effects of the two cracks.

From both scenarios of the state of stress presented in Figure 5 and Figure 6 it is shown in the fourth image, obtained for Δ*c* < 4 mm that the closer the damage the bigger the effect on the frequency drop will be. The biggest deviation from the results obtained by superposing the effects is found for the cracks located at the same distance from the fixed end Δ*c* = 0 mm. After passing the position *c_F_ =* 210 mm, the bottom crack is moving away from the upper crack and the frequency drop decreases consequently.

## 5. Conclusions

In this paper we analyze the behavior of a beam subjected to transverse cracks and the possibility to model this beam as a beam with uniform cross-section but with different geometrical or physical-mechanical parameters. We found out that, for the particular case where the crack is located on the slice where the beam is subjected to the highest bending moment, the developed EHB can be used to model the behavior of the cracked beam in terms of eigenfrequencies. This model consists of a healthy beam with a different thickness *h_eq_* or mass density *ρ_eq_* and is valid for any boundary conditions imposed to the beam. If the damage is located in other slices, the thickness *h_eq_* or mass density *ρ_eq_* have to be adjusted with a correction coefficient which consider the local value of the bending moment (or curvature) for the targeted vibration mode. The advantage of using healthy beams to model the behavior of cracked beams is the simplicity. Comparing the results obtained by FEM simulation and by calculus involving the ML-EHB, we found errors less than 0.005%.

We used the ML-EHB to model the beam with two cracks too, and found very small differences between the results obtained in this way and from FEM simulation if the cracks are far enough from each other. This means the superposition principle can be applied in these cases.

For a small distance between the two cracks located at the same beam face, we achieved a lower frequency as the superposition predict. This happens because the effects of the cracks interfere and the beam segment between the cracks behaves as a single crack. Obviously, if the distance between the cracks is infinitesimally small, these have the effect of a single crack.

Another rule was found for closely located cracks that affect opposite faces. In this case, the frequency decrease due to the two cracks is bigger as that calculated involving the superposition principle. This is because the two cracks can be assimilated with a crack which has as depth the sum of depths of the two cracks, and it is known that the beam frequency decrease exponentially with the depth of the crack.

In order to apply the method, it is necessary to know the mode shapes of the structure’s elements. We tested the method on slender beams, whose behavior can be described with the Euler–Bernoulli model.

The proposed method is also suitable for single cracks on sandwich type beams. Our further research will also address such beams with various types of defects. The weakness of the method, as with all vibration-based procedures, is the need to use very precise modal parameters.

## Figures and Tables

**Figure 1 sensors-21-05215-f001:**
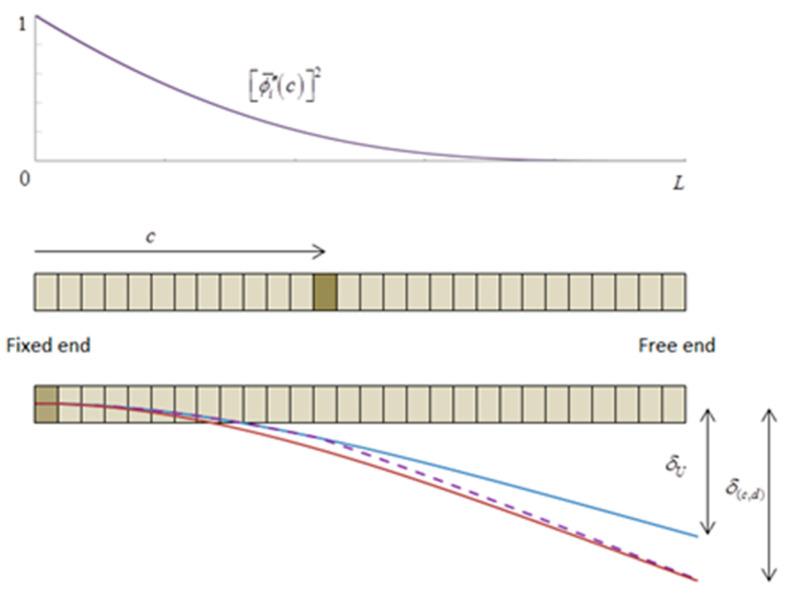
The model for the beam with a crack located at distance c and the model attaining the same deflection with the crack at the fixed end.

**Figure 2 sensors-21-05215-f002:**
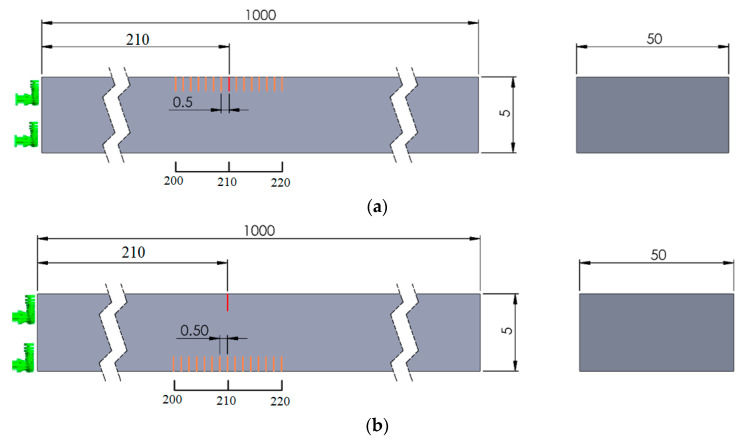
(**a**) Mobile crack on the upper face of the beam; (**b**) Mobile crack on the lower face of the beam.

**Figure 3 sensors-21-05215-f003:**
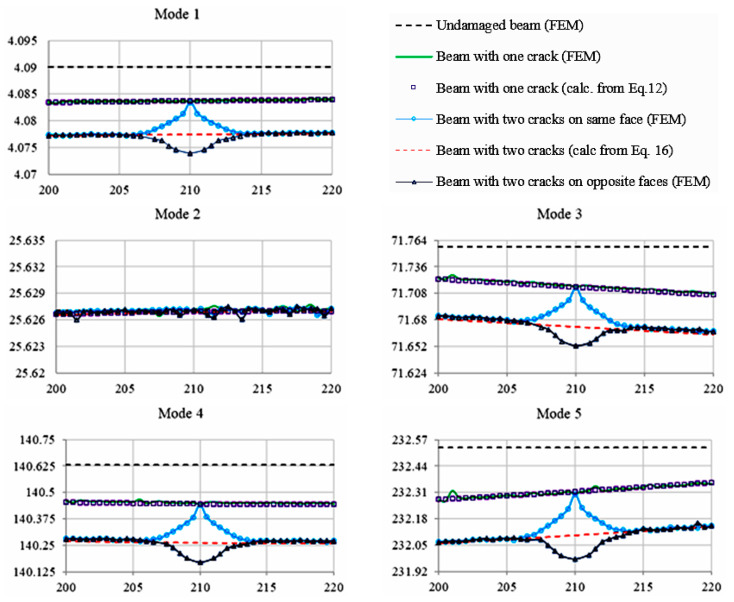
Frequency evolution with the crack location in the case of one and two cracks.

**Figure 4 sensors-21-05215-f004:**
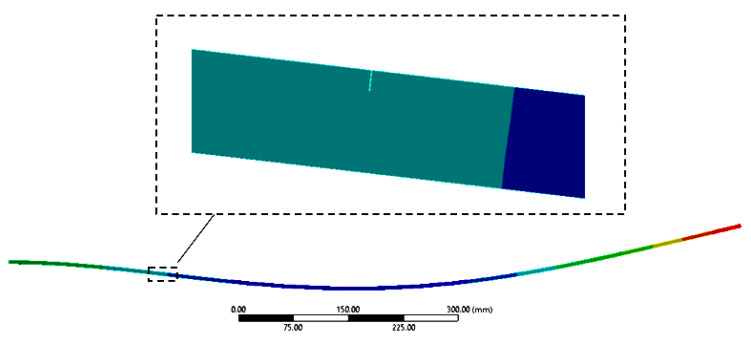
Zoom on a crack positioned at an inflection point.

**Figure 5 sensors-21-05215-f005:**
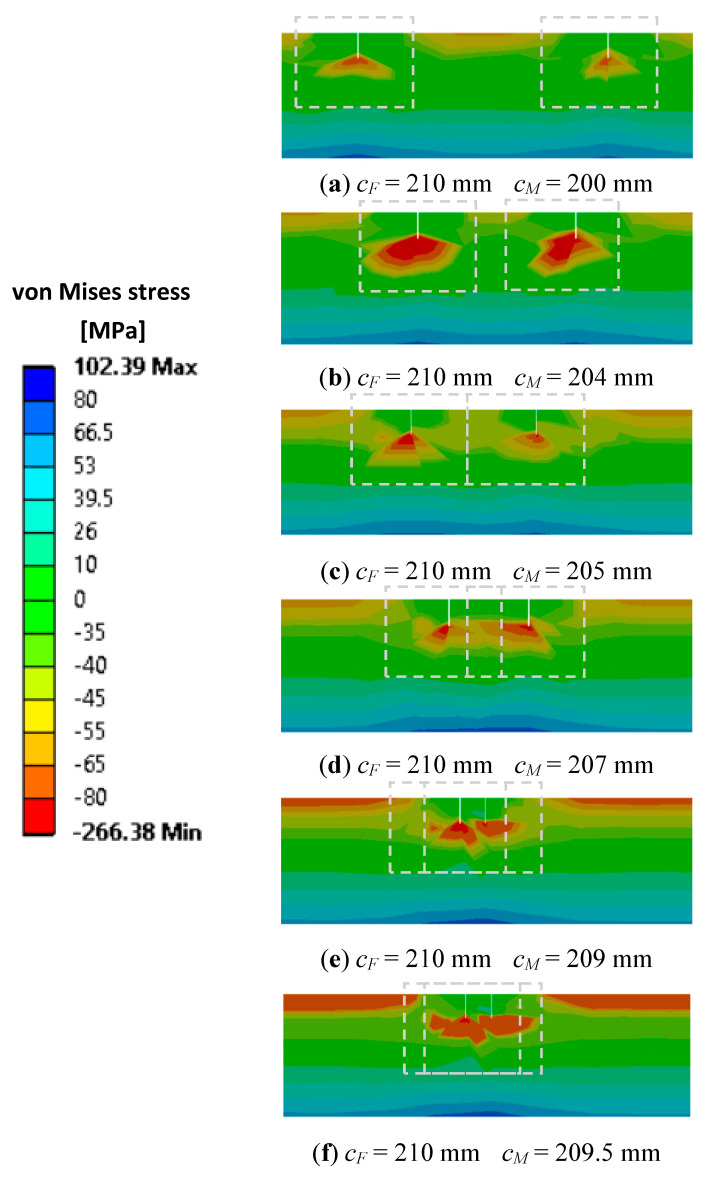
Interfering of the stress state disturbances for the two cracks positioned on the same face.

**Figure 6 sensors-21-05215-f006:**
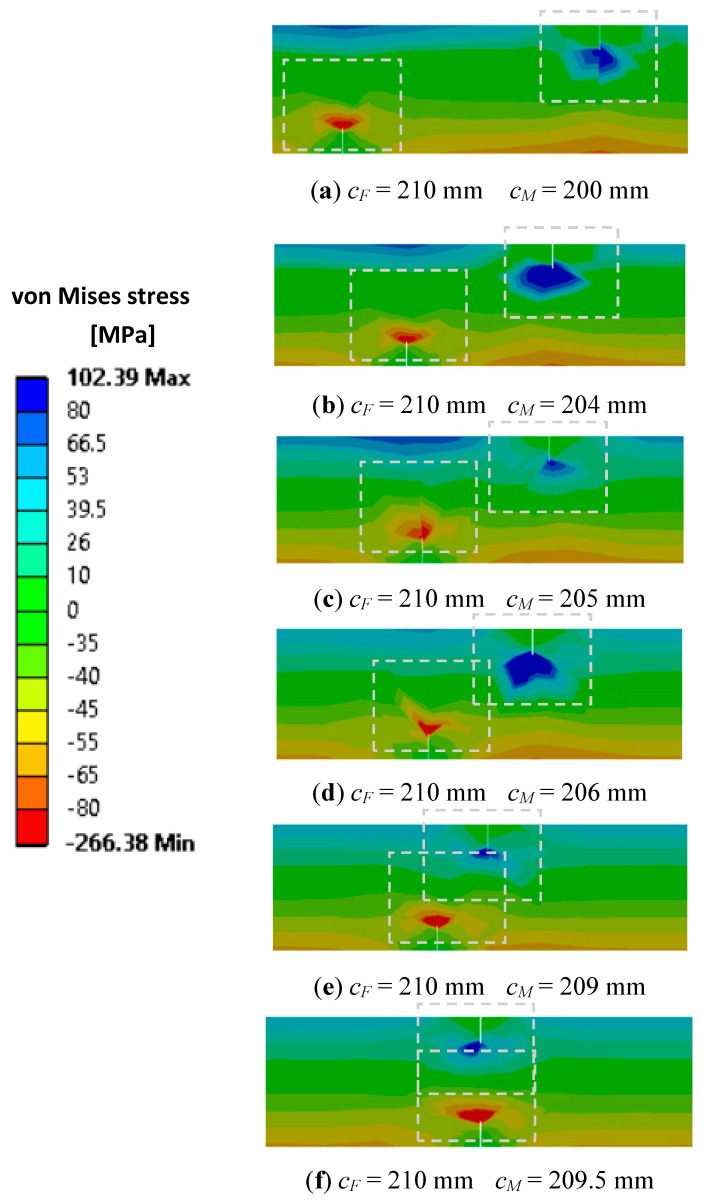
Interfering of the stress state disturbances for the two cracks positioned on opposite faces.

**Table 1 sensors-21-05215-t001:** Physical, mechanical and geometrical properties of the beam.

Length *L* (mm)	Width *b* (mm)	Thickness *h* (mm)	Mass Density *ρ* (kg/m^3^)	Young Modulus *E* (N/m^2^)	Poisson Ratio *ν* (-)
1000	50	5	7850	2 × 10^11^	0.3

**Table 2 sensors-21-05215-t002:** Frequency drop for the beam with one crack obtained by simulation.

Mode No.	$\Delta f_{i}^{FEM}(100,1)$ [Hz]	$\Delta f_{i}^{FEM}(210,1)$ [Hz]	$\Delta f_{i}^{FEM}(400,1)$ [Hz]	$\Delta f_{i}^{FEM}(600,1)$ [Hz]	$\Delta f_{i}^{FEM}(700,1)$ [Hz]
1	0.002249	0.001516	0.000636	0.000147	0.000049
2	0.000819	0	0.001015	0.001366	0.000858
3	0.000167	0.000571	0.000683	0.000822	0.001742
4	0	0.001279	0.000284	0.000284	0.000569
5	0.000258	0.000946	0.001462	0.001462	0.000172
6	0.000662	0.000115	0.000029	0.000029	0.001353

**Table 3 sensors-21-05215-t003:** Frequencies obtained from direct simulation and by superposing the crack effect.

Mode No.	Freq. *c_F_* = 210 mm *c_M_*_1*F*_ = 100 mm		Freq. *c_F_* = 210 mm *c_M_*_2*F*_ = 400 mm		Freq. *c_F_* = 210 mm *c_M_*_3*F*_ = 600 mm		Freq. *c_F_* = 210 mm *c_M_*_4*F*_ = 700 mm	
	FEM	Superp.	FEM	Superp.	FEM	Superp.	FEM	Superp.
1	4.0746	4.0746	4.0812	4.0812	4.0832	4.0832	4.0835	4.0836
2	25.606	25.606	25.601	25.601	25.591	25.592	25.606	25.605
3	71.704	71.704	71.668	71.667	71.657	71.657	71.593	71.591
4	140.45	140.45	140.41	140.41	140.41	140.41	140.37	140.37
5	232.25	232.25	231.97	231.97	231.98	231.97	232.28	232.27
6	347.18	347.19	347.40	347.41	347.41	347.41	346.95	346.95

**Table 4 sensors-21-05215-t004:** Squared modal curvatures and correction coefficients for crack positions $c_{M}\in \left[200,220\right]$ [mm].

Crack Location *c_M_* [mm]	Squared Modal Curvature					Correction Coefficient				
	Mode Number					Mode Number				
	1	2	3	4	5	1	2	3	4	5
200	0.526318	0.004905	0.155925	0.413497	0.360538	0.99840	0.99999	0.99953	0.99874	0.99891
202	0.522376	0.003779	0.163434	0.417974	0.350399	0.99841	0.99999	0.99950	0.99873	0.99894
204	0.51845	0.002803	0.170995	0.422	0.3398	0.99843	0.99999	0.99948	0.99872	0.99897
206	0.514541	0.001974	0.178598	0.425568	0.328777	0.99844	0.99999	0.99946	0.99871	0.99900
208	0.510647	0.001293	0.186233	0.428673	0.317371	0.99845	1.00000	0.99943	0.99870	0.99904
210	0.50677	0.000756	0.193892	0.431308	0.305621	0.99846	1.00000	0.99941	0.99869	0.99907
212	0.502909	0.000364	0.201564	0.43347	0.293569	0.99847	1.00000	0.99939	0.99868	0.99911
214	0.499064	0.000114	0.209241	0.435155	0.281257	0.99848	1.00000	0.99936	0.99868	0.99915
216	0.495235	0.00000537	0.216913	0.436362	0.268729	0.99850	1.00000	0.99934	0.99868	0.99918
218	0.491423	0.0000362	0.224571	0.437088	0.256029	0.99851	1.00000	0.99932	0.99867	0.99922
220	0.487626	0.000205	0.232205	0.437333	0.243201	0.99852	1.00000	0.99930	0.99867	0.99926

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
