# Peer review of "Damage Detection on a Beam with Multiple Cracks: A Simplified Method Based on Relative Frequency Shifts†"

_sensors, 2021, doi:10.3390/s21155215_

Round 1

Reviewer 1 Report

This paper has some innovation The damage indicator who developed was based on
the relative frequency shifts (RFS) Identifying cracks in the incipient state is essential
to prevent the failure of engineering structures. There have some advice to discussion.
1. The effect of reinforcement ratio is not considered in the numerical simulation
and whether the reinforced concrete beam with this method is reasonable?
2. What material is the cantilever made of?
3. Is this method suitable for cantilever beams of anisotropic materials?

4.  The effect of reinforcement ratio is not considered in the numerical simulation,and whether the reinforced concrete beam with this method is reasonable?

Author Response

se attached file

Reviewer 2 Report

The manuscript describes a damage detection method on a beam with multiple cracks based on relative frequency shifts outlier detection technique. The topic is of general interests to the reader of the journal of Sensors and the paper is worthy of publication in the journal if the following concerns are addressed:

  1. The literature review is very poor. Most of the literature is from 2019 or earlier and lacks cutting-edge technologies.
  1. Are there any special requirements on the crack or the beam that the technique can apply?
  1. There is lack of comparison with the literature. In particular, it is essential that the authors demonstrate the quality or efficiency of their results, compared to well-established methods.
  1. The figure is not well expressed and explained. For example, figure 4 is not explained, and figure 5 lacks units of stress.
  1. Lack of organization in data analysis, particularly from line 366-371. Whether the two cracks represented by the red line is on same face or opposite faces of the beam, and the blue line or the triangle-line lack the control group.
  1. The conclusion part should be written more comprehensively. It also has to be extended with the weak points of the provided method and further studies.

Reviewer 3 Report

The Authors have presented a method for recognizing damage of a beam based on analysis of eigenfrequencies. The considered damages are composed of multiple cracks. As suggested in the paper, the advantage of the method lies in recognizing the closely located cracks. The paper is interesting. However, there are several essential weaknesses:

The Authors have stated that "We can accurately locate multiple cracks and estimate their severity trough experiments". However, there are no results presented in this paper that would confirm the above statement straightforwardly. My suggestion is to add the results regarding the crack localization error (accuracy) and error or accuracy of the crack severity estimation. These results have to be compared for the proposed approach and the previous methods (e.g., the method based on superposition).

Moreover, the results shown in this paper are limited to two cracks, while the method is designed for multiple cracks. Therefore, the method's applicability should be discussed and demonstrated in cases when the number of cracks is higher than two. The Authors should also discuss the dependency between the number of cracks, the computational complexity of the method, and the size of the database. Limitations of the method have to be clearly indicated. For instance: what is the maximum number of cracks that can be recognized in real-world applications? It is also not clear if the method recognizes the number of cracks or that number has to be known in advance.

How can the beam damage be recognized using the proposed method in real-world conditions? Would you please describe the necessary sensors and measurement procedure?

In Section1, the main contributions of the paper should be better explained.

In my opinion, the previous conference paper of the Authors entitled “Study regarding the effect of crack branching on the eigenfrequencies of beams” should be cited in the main text with a discussion of the advances made in the manuscript submitted to the Sensors journal. 

Line 436: “Comparing the results obtained by FEM simulation and by calculus involving the ML-EHB fit, the errors being less than 0.2%.”. Please provide extended experimental data and add a discussion of the error analysis. How this error was determined?

Line 286: Please explain the "intelligent optimization methods" and how they can be used.

Line 39: "trough experiments" -> through experiments

Round 2

Reviewer 3 Report

The authors have improved the paper by considering my suggestions satisfactory. I believe the paper can be accepted for publication.